# Epithelial to Mesenchymal Transition as Mechanism of Progression of Pancreatic Cancer: From Mice to Men

**DOI:** 10.3390/cancers14235797

**Published:** 2022-11-24

**Authors:** Luana Greco, Federica Rubbino, Luigi Laghi

**Affiliations:** 1Laboratory of Molecular Gastroenterology, IRCCS Humanitas Research Hospital, Via Manzoni 56, 20089 Rozzano, Italy; 2Department of Medicine and Surgery, University of Parma, 43126 Parma, Italy

**Keywords:** EMT, models, translational research, personalized medicine, pancreatic cancer

## Abstract

**Simple Summary:**

The progression of pancreatic cancer (PDAC) involves a series of events transforming the phenotype of cancer cells, namely epithelial-to-mesenchymal transition (EMT), which contributes to their invasiveness and spread. Although basic science studies, by in vitro and in vivo animal models strongly support the occurrence of EMT in PDAC, it remains perceived as something heretical in a clinical perspective. To turn this suspended perception, more translational data are needed, consolidating the notion of EMT as a hallmark of human PDAC. Clearly, the network of mechanisms involved its timing and regulation also requires further research, such as those aspects of EMT cancer cell at the intersection with stemness. The translational improvement provided by the identification of EMT markers suitable for deciphering the aggressive behavior of PDAC could eventually modify the clinical scenario, possibly contributing to the advancement in diagnosis and monitoring of its evolution and responsiveness to treatments.

**Abstract:**

Owed to its aggressive yet subtle nature, pancreatic cancer remains unnoticed till an advanced stage so that in most cases the diagnosis is made when the cancer has already spread to other organs with deadly efficiency. The progression from primary tumor to metastasis involves an intricate cascade of events comprising the pleiotropic process of epithelial to mesenchymal transition (EMT) facilitating cancer spread. The elucidation of this pivotal phenotypic change in cancer cell morphology, initially heretic, moved from basic studies dissecting the progression of pancreatic cancer in animal models to move towards human disease, although no clinical translation of the concept emerged yet. Despite this transition, a full-blown mesenchymal phenotype may not be accomplished; rather, the plasticity of the program and its dependency on heterotopic signals implies a series of fluctuating modifications of cancer cells encompassing mesenchymal and epithelial features. Despite the evidence supporting the activation of EMT and MET during cancer progression, our understanding of the relationship between tumor microenvironment and EMT is not yet mature for a clinical application. In this review, we attempt to resume the knowledge on EMT and pancreatic cancer, aiming to include the EMT among the hallmarks of cancer that could potentially modify our clinical thinking with the purpose of filling the gap between the results pursued in basic research by animal models and those achieved in translational research by surrogate biomarkers, as well as their application for prognostic and predictive purposes.

## 1. Complexity of Metastatic Cascade, Embracing Stromal and Mesenchymal Traits, and the Importance of Its Application to Human Pancreatic Cancer

Oncogenic and tumor suppressor derangements drive cell transformations and tumor progression; however, they are likely not sufficient *per se* for metastatic competence. This concept has been first depicted for colorectal cancer, a mainstay model for gene damage and tumor progression. While perfecting this model, some of the authors initially involved in its drawing acknowledged that the waves of clonal expansions from early to advanced carcinoma and then to metastasis were not associated with known genetic alterations [1]. Since then, several transgenic mouse models and clinical observations underpinned that disseminated neoplastic cells do not invariably develop distant metastasis [2,3,4,5,6,7]. 

Among cancer hallmarks, those related to metastasis development remain complex to explore and unravel, and the knowledge on local tumor progression developed in the last three decades is not sufficient to explain it. According to the concept of metastatic cascade, cancer cells should acquire additional abilities to overcome the barriers against metastatic spread. In this context, progression toward metastasis includes the contribution of host cells pushing toward the enhancement of cancer hallmarks [8,9,10,11] Thus, besides genetic drivers of clonal evolution, other players are involved in metastasis development. Altogether, host cells of the peritumoral stroma contribute to the complex entity referred to as “tumor microenvironment” [12,13]. It has been implicated in the regulation of cell growth, in affecting metastatic potential and possibly the location of secondary lesions, as well as the responsiveness to therapy. Although such stromal cells are not properly malignant, they play an essential role in supporting the survival and growth of cancer cells, and are an attractive target for anticancer treatments [14].

Considering the modifications of the cells composing the complex array of tumor microenvironment, it should not be disregarded that epithelial cells acquiring plasticity can also assume the features of other cells. Such changes along EMT implicate cellular de-differentiation and an increase in motility following the loss of cell–cell adhesion [15]. Ideally, in the reverse change referred to as mesenchymal–epithelial transition (MET), cells lose migratory freedom, re-expressing junction complexes and again adopting apicobasal polarity [16]. These metamorphoses are involved in developmental and physiological processes, as well as in diseases and ensuing damage response programs. In solid tumors, the activation of EMT occurs at the invasive front by a convergence of signals and cross-talks between the tumor nests and the microenvironment, so that cancer cells that underwent EMT become able to invade the stroma and blood vessels [17]. Studies also indicate that metastatic cancer cells which have presumably undergone EMT may exhibit a cancer stem cell (CSC) phenotype. For instance, in pancreatic tumors, CD133+ mesenchymal-like cells that also express the CXCR4 chemokine receptor are found predominantly at the invasive front of tumors where they may be primed for metastatic spread [18]. Poorly differentiated and aggressive cancers show a gene expression signature related to that of embryonic stem cells [19], supporting the existence of relationship between EMT and CSC-like phenotype, which may be prerequisites for cancer cell metastasis in which the major driving force is the TGF-β signaling pathway [20,21]. In addition, the state of tumor may contribute to its drug resistance. For instance, an increasing body of literature suggests that reversible epigenetic changes emerging during acquired drug resistance reflect changes in the differentiation state of the tumor, which is likely to reflect EMT [22,23]. Despite the evidence supporting the exploitation of EMT and MET during cancer progression [24], our understanding of the relationship between tumor microenvironment and EMT is not yet mature and needs additional data from basic, translational, and clinical sciences for clarification. 

## 2. Models to Study EMT in Pancreatic Cancer

High PDAC lethality, together with its inadequate response to surgical and medical treatments, makes it important to recapitulate and unravel its pathological and molecular hallmarks taking advantage of preclinical models, to envision new perspectives for interfering with its so far unrestrainable progression [25]. In this respect, the genetic damage of PDAC is characteristically simple in its components. It chiefly includes *KRAS* activation, which occurs early and individually already in precursor lesions [26] (i.e., Pancreatic Intraepithelial Neoplasia, PanIN), additionally comprising the inactivation of *TP53* as well as of *CDNK2A* and *SMAD4*. Besides this almost monotonous background in PDAC cells, associated genomic aberrations through the genome can help stratify PDAC according to the structural variants per genome (stable, scattered, locally rearranged, and unstable genomes) [27], although instability due to DNA mismatch repair defects such as observed in colorectal cancer is rare in PDAC [28,29] and does not allow a relevant rate of exploitation for therapeutic actions [30,31]. Lately, several authors have employed transcriptomic profiles to categorize PDAC. This approach revealed several subtypes and afterwards led to the attempt to harmonize them. One of the inherent difficulties of this approach is to distinguish stromal signatures from truly neoplastic ones. Furthermore, stroma composition may oppositely affect the outcome depending of molecular subtype [32]. Interestingly, a quasi-mesenchymal portrait of PDAC first detected by Collison et al. has been subsequently categorized as basal-like (Moffitt [33]) or squamous (Bailey [34]) or eventually as desmoplastic and stroma activated (Puleo [32]). On the other hand, immune response mounted against PDAC may also explain the differences in disease progression [32,35], although sneaky progressive behavior of PDAC remains largely elusive. However, the hallmark(s) of the involvement of EMT in human PDAC remains to be identified yet, as opposed to the above reported genetic derangements. 

Unlike the results of the phylotranscriptomic of human PDAC, data from animal models point to the expression of mesenchymal markers as an inherent feature of pancreatic cancer. Such expression is pronounced in nano- and micro-metastatic lesions, while the re-emergence of epithelial features takes place in macro-metastatic lesions. Furthermore, such dichotomy between strong evidence supporting EMT in PDAC animal models as compared to similar data in humans is wide, with some remarkable exception [5].

In this review, we attempt to resume the pivotal yet largely unraveled role of EMT in the progression of pancreatic cancer (i.e., pancreatic ductal adenocarcinoma, PDAC) also in relation to aspects not yet clarified in humans. Animal models strongly point to EMT as a key feature for PDAC dissemination, and we will briefly cover the preclinical models which are listed below and schematized in Figure 1.

### 2.1. Genetically Engineered Mouse Models (GEMMs)

The development of genetically engineered mouse models, mimicking the main genetic events occurring in human PDAC, enhanced our notions on this neoplastic disease. 

KC (*Kras* ^LSL.G12D/+;^ Pdx-1-Cre) and KPC (*Kras* ^LSL.G12D/+^; *Trp53* ^R172H/+^; Pdx-1-Cre) animals are widely used for mimicking the occurrence of PDAC. The KC model, consisting of knock-in strain of LSL-*Kras* ^G12D^ and transgenic strain of Pdx1-Cre recapitulates PDAC with developmental attributes similar to its human counterpart [36]. The model was first developed by Tuveson and colleagues, thereafter in 2005 *Kras* mutation was coupled with *Trp53* inactivation [37] (the ortholog of commonly deranged tumor suppressor *TP53*) [38], creating the KPC model [39]. This Cre-lox-based mouse model developing PDAC with an almost 100% penetrance improved the study of the fate of pancreatic epithelial cells during tumor progression [40].

KPC and KC mouse models differ in terms of their inferences: KC model was the first one to display the progression from pre-cancerous lesions (i.e., PanIN) up to metastatic tumors. It was also used to assess the core function of *Notch* in PDAC development [41], showing that the removal of Notch1 in the KC mouse model sped up PanIN development [42].

Compared to KC model, the KPC GEMM invariably develops PanIN lesions within 10 weeks of age, which rapidly progress to PDAC within 16 weeks of age. The median survival of these mice is 5 months, with the occurrence of malignancies and metastases [39]. Rhim and colleagues, by using in vivo lineage tracing, captured EMT process very early during neoplastic progression [2]: epithelial cells can migrate into the stroma, entry into the bloodstream, and reach the liver already at pre-invasive tumor stage of PanIN.

Primary tumors and metastases in KIC model (*LSL-Kras*^G12D^; *Cdkn2a*^lox/lox^; *p48*^Cre^), when treated with drugs targeting stromal Tgfβr2, showed EMT inhibition by increasing the expression of tumoral E-cadherin coupled with Vimentin reduction, but did not rule out the promotion of MET [43].

However, a recent study by Lan et al. demonstrated that overexpression of the soluble factor Grem1 caused an almost complete “epithelialization” of highly mesenchymal PDAC, indicating that high GREM1 activity is sufficient to revert EMT process [44]. In addition, high expression of *Grem1* in mesenchymal PDAC cells was coupled with reduced expression of *Snail* and *Slug* in epithelial compartment, showing the involvement of paracrine signaling in the maintenance of PDAC cellular heterogeneity [44]. 

Altogether, the evidence supports a model for pancreatic cancer progression in which the dissemination and the seeding to distant organs occurs before, and in parallel to, primary tumor formation. One other GEMM model, KPfC (*Kras*^LSL−G12D/+^; *Trp53*^fl/fl^; *Pdx1*^Cre/+^), showed two distinct cancer cell populations, one epithelial and one mesenchymal, sharing gene alterations with KIC model. This finding highlights that under the same oncogenic *Kras* driver, different secondary mutations can lead to alterations in signaling pathways driving PDAC progression, supporting a model in which mesenchymal features of cancer cells are acquired later in the disease process [45].

The inherently plastic and transient nature of the EMT makes it difficult to obtain a clear fingerprinting, and molecular tracer can somehow be elusive. Studies from Kalluri and coll. revealed a suppressed EMT program in KPC GEMMs lacking *Twist* or *Snail* [46]. By a dual-recombinase system-driven model in combination with lineage tracing of EMT program (αSMA-Cre and Fsp1-Cre), they showed the reduction of the EMT program in primary PDAC altogether with an enhanced proliferation. The authors argued that EMT program is dispensable for initiation and progression of PDAC. Furthermore, as KPC mice with knocked down *Twist* showed both circulating tumor cells as well metastasis formation, they also inferred that the suppression of EMT in PADC does not alter the rate of dissemination. These conclusions were later challenged by the criticisms that αSMA, the marker employed to trace cancer cells with mesenchymal features, is rarely induced upon EMT activation in KPC model. Furthermore, an additional KPC model in which *Zeb1* has been deleted developed PDAC anyway, although with better tumor differentiation and retained *Twist1* expression despite a reduction of *Zeb2*, Slug and Snail expression. Pancreatic tumors in these mice showed reduced stemness as well as lower propensity for the development of metastases, less tumorigenic and colonization capacities [46]. However, critical reappraisal of these data points to functional differences between EMT-TFs in tumor progression [6], underscoring the relevance of *Zeb1* for plasticity and metastasis of pancreatic cancer. As to tumor genetics, the differences between the effects of *Zeb1* depletion in KPC and KC mice underscore its crucial role for the formation of *K-ras* driven PanINs, and that mutated *p53* acts as an accelerator towards metastasis. 

However, the argument concerning the hierarchy of EMT transcription factors as to PDAC progression was far from being fixed. A relevant paper joined the evaluation of EMT profiles taking advantage of the revision of previous work in humans [47] with the analysis of GEMM bearing purposed gene deletions to explore the feasibility of dissecting tumor cell content according to the spectrum of EMT signature [48] and evaluate the progression towards metastasis. Again, *Zeb1* deletion led to liver metastasis with stabilized epithelial morphology of PDAC cancer cells, coupled with their collective migration. While the data showed that inhibition of EMT does not impede metastasis in GEMM thus leaving open the question as to the effective role of EMT states in metastasis development, the parallelism between human and mouse transition from epithelial to mesenchymal states of cancer cells was unquestionable [5]. Accordingly, we should acknowledge that truly epithelial PDAC cells are only a fraction (roughly 1/3) of the total tumor burden, the remaining fractions being either in intermediate E-M states or closely mesenchymal. These data are of interest also in light of the alternative ways to switch-on EMT, as shown by Aiello and coll., who showed that EMT activation can be achieved by a different mechanism of protein internalization rather than by transcriptional repression only, ensuing in intermediate EM phenotypes and in migration of clusters of cancer cells [49]. These findings support a dual mode for undertaking EM, achieving complete or partial transition (the latter by CDH1 internalization) which should be kept in mind when interpreting models pursued by selective abrogation of specific EMT transcription factors. Again, a parallelism with human PDAC types was brought up by the authors, namely the complete EMT resembling quasi-mesenchymal tumors and squamous ones. 

Deepening the understanding of the role of *K-ras* mutational pattern, in 2018, Mueller and colleagues demonstrated that oncogenic dosage-variation has a critical role in PDAC differentiation [50]. Notably, an increased rate of *KRAS* mutations in human PDAC precursor was found, which was linked with early tumorigenesis as well with metastatization. By characterizing PDAC cells expressing *Kras^G12D^* conditionally in the pancreas (PK mice) and by the comparison with results from WES (whole-exome sequencing) of human PDAC, they found a similar mutation pattern among species but recurrently altered genes were infrequent in mice. Four states were identified in *Kras^MUT^* gene dosage: focal gain, arm-level gain, neutral loss of wild type, and no change. Altogether, two-thirds of PDAC showed an increased *Kras* dosage due to allelic imbalance, and those cancers had increased metastatic potential. Additionally, the most frequent mPDAC deletion occurs at *Cdkn2a* and/or adjacent regions, with higher Kras^G12D^ expression. Accordingly, in micro-dissected human PDAC data sets, *KRAS^MUT^* variant allele frequencies were higher in *CDKN2A*^∆HOM^ than in *CDKN2A*^∆HET/WT^ tumors [50]. Furthermore, the mesenchymal phenotype was associated with an enhanced expression of *Kras^MUT^*, although such a strong EMT was only described in mice but not in human cells yet, therein *KRAS^MUT^* overexpression led to vimentin overexpression and CDH1 repression. Similar findings were recapitulated in undifferentiated human PDACs, marked by the upregulation of EMT linked genes.

In summary, the role of EMT-TFs in PDAC has been focused but it is still controversial. For instance, genetic depletion of *Snail* or *Twist* in the KPC model has no effect on the development of metastasis [46] but genetic ablation of *Zeb1* profoundly reduced the metastatic capacity of PDAC tumors [6]. However, the way in which nutrient stress contributes to EMT, metastasis, and the regulation of EMT-TFs in PDAC has been poorly explored. Recouvreux and colleagues showed that glutamine depletion leads to the activation of the EMT transcriptional program, identifying a novel mechanism of EMT induction in human and mouse model of PDAC, orchestrated by *Slug* upon glutamine starvation; then, suppression of *Slug* in vivo profoundly abrogated the metastatic capacity of PDAC cells [51]. 

Notwithstanding copious knowledge benefits, these models are expensive, time-consuming and limited to oncogenic events introduced in a monotonic mouse genetic background, which is unlikely to properly reflect the complexity of human PDAC with respect to genetic heterogeneity of humans. In recent years, it is becoming clear that it is also necessary to functionally explore cell–cell interactions, adding surrounding stromal cells to recapitulate dynamic PDAC behavior [52,53].

### 2.2. Heterotypic Human Cell Co-Culture Models

A common way to mimic PDAC microenvironment, as consequence the presence of EMT process, is by co-culturing, with or without physical contact, PDAC cells and CAFs in monolayers. In the first case, cells are mixed allowing physical interactions, while for the second case cells are cultured in separate chambers (such as trans-well system) and cell–cell communication occurs only through diffusion of soluble factors. Ligorio et al. [52] demonstrated by single-cell RNA-sequencing that CAF secreted factors can modify PDAC features. Particularly, the acquisition of EMT phenotype is driven by TGF-β secreted from CAFs, which is enriched within the stroma. However, even though this simple method can mimic some aspects of PDAC tumors, it can be further improved by inclusion of matrices such as collagen, laminin, hyaluronan, as well as by the introduction of hypoxic and low serum conditions [54].

### 2.3. Spheroids

In recent models, PDAC cells are embedded three-dimensionally as spheroids altogether with stromal cells in a milieu resembling blood stream. In this way, cells produce more matrices and develop chemoresistance compared to when they are grown as monolayers [55]. PDAC cells with intact TGF-β signaling machinery adopt a pronounced EMT phenotype when cultured as spheroids compared to monolayer culture. Immunocytochemically, different results were observed for epithelial and pancreatic cancer cells grown in 2D and 3D culture systems [56].

It has been shown that arrangement in 3D structures modifies the expression of EMT markers in PDAC cells. Namely N-cadherin was more evident in 3D spheroids than in 2D monolayers, while podoplanin was similarly expressed in both conditions, altogether with the maintenance of CDH1/beta-catenin complex at cell boundaries, altogether sup-porting collective cell migration [57].

In 3D culture, epithelial cells had high E-cadherin and CA19.9 expressions whereas Vimentin was low, exhibiting a round-like appearance encircled by flat cells, conversely of PDAC cells which formed grape-like spheres. In both 2D monolayer and 3D cultures, cell lines show variable expression of epithelial markers and carcinoembryonic antigen (CEA) tumor marker. Differently, Vimentin was localized in most pancreatic cancer cells but only in a small number of adherent epithelial ones, while it was absent in cell spheres. The expression of other markers only in epithelial cells and in their spheroids, such as the normal pancreatic ductal marker Cytokeratin 7 (CK7), the tumor marker CA19-9, and other normal pancreatic acinar cell marker i.e. E-cadherin and trypsin, suggests that the epithelial cells can differentiate to pancreatic ductal or acinar cells. Surprisingly, in 3D culture, flat cells exclusively showed Ki-67 immunoreactivity suggesting that only these cells proliferate in the spheroid resembling the proliferating zone of normal epithelial tissues. They also may differentiate to cancer cells inside the spheres. Unlike flat cells, in PDAC cell spheroids Ki-67 was irregularly expressed. In addition, loss of CK7 and trypsin expression in pancreatic cancer cells spheres suggest both morphological and functional de-differentiation, while the expression of Vimentin suggests their mesenchymal phenotype [56]. These features indicate that three-dimensional culture better recapitulates PDAC cells functions and characteristics.

Recently, an engineered tumor models to recapitulate a 3D microenvironment to study the TME and PDAC was developed. Bradney et al. proposed a mechanism of EMT and local invasion caused by the interaction between heterogeneous cancer cell populations. They created a tumor 3D biomimetic model named ductal tumor microenvironment-on-chip (dT-MOC) which permits analysis and experimentation on the epithelial–mesenchymal transition (EMT) and local invasion with intratumoral heterogeneity. This model used a murine model pancreatic cancer cell (PCC) derived from genetically engineered mouse models which carries key driver mutations of human PDAC including *KRAS*, *CDKN2A*, and *TP53* mutations. A complex interaction even between cancer cells was showed to lead to a more aggressive and invasive nature of the pancreatic cell under treatment with TGF-β. This suggests that cancer cells with mesenchymal features can induce EMT and enhance local invasion of other PCCs [58].

### 2.4. Organoids

The latest technology in the field of PDAC culture is referred to as organoids, introduced and developed by Tuveson and Clevers groups; it can preserve cellular polarity and tissue architecture starting from human biopsies [59]. Boj and colleagues were able to isolate ductal organoids from murine primary tumors and metastases from both KC and KPC mice, characterized by a prominent stromal response, trait often absent in monolayer cell lines. A close resemblance of the organoid transplantation models with autochthonous PDAC was observed. Additionally, was highlighted a low vascular density and high vessel-to-tumor distance, in contrast to transplanted 2D cell lines [60]. Finally, they also demonstrated that mouse organoid systems are accurate for studying PDAC progression by selecting upregulated genes that were validate by immunohistochemistry and immunofluorescence in human tissues. Even though PDAC organoids represent currently the most powerful in vitro model, their isolation, establishment, and expansion require weeks to months, specific growth factors, and highly trained personnel to maintain the intra-tumoral heterogeneity, limiting the adoption of this technique.

## 3. EMT and Human Pancreatic Cancer 

### 3.1. A Potential Route from Translational Studies to Personalized Medicine?

Several master transcription factors (TFs) can activate the EMT [7,15]. Among these, *TWIST1* plays an essential role both in normal development and cancer metastasis [61]. In other gastrointestinal cancers such as colon cancer, the stromal expression of *TWIST1* has been associated with worse prognosis, even with neoplastic features such as trisomy, and *TWIST1* elevated mRNA circulating levels have been detected in the blood of colon cancer patients [62]. 

Among various mechanisms activating *TWIST1*, little is known concerning the significance of *TWIST1* methylation in human PDAC, in which it appears to be frequently methylated [63]. It is known that the hypermethylation of DNA in promoter of CpG islands results in the transcriptional silencing of cancer-related genes. In this case, the methylation of the proximal region of *TWIST1* promoter does not relate to its expression but to the expression of neighboring genes, i.e. *HDAC9* and *N-TWIST* [64]. Other EMT inducers, such as *SNAIL* and *SLUG*, are expressed in pancreatic cancer but not in normal tissue, suggesting their role in the progression of human pancreatic tumors [65]. Notably, cells in premalignant pancreatic lesions (such as PanIN or IPMN) can also undergo EMT [66,67]. 

Integrated genomic, transcriptomic, and proteomic profiling identified significantly different scores for EMT activation. Undifferentiated human PDAC are characterized by a strong upregulation of genes enriched for EMT and RAS downstream signaling pathways, combined with reduced expression of genes involved in epithelial or squamous differentiation [50,68]. Furthermore, a favorable prognostic subset characterized by low EMT signature [68] confirmed that the activation of this program, consistent with the features proposed by Thiery and coll., correlates with previous classifiers that bring a worse patient outcome [48]. Specifically, they proposed a generic EMT score as promising tool to estimate EMT phenotypes by transcriptomic profiles in different cancer types. Such profiling could help in assessing cancer progression as well drug responsiveness. Despite the great potential of the proposed algorithm, the correlations of EMT with poorer survival and relapse were not without exception observed in all cancers, suggesting the potential for the stratification of patients by EMT score [48]. However, it has been shown, both for immunohistochemical and gene expression analysis, that the great majority of human metastatic PDACs lesions are purely epithelial [69,70,71], even exhibiting more epithelial features if compared to primary tumor [53]. The prognosis of patients affected with neoplasia, sometimes even with small primary tumor, mainly depends on the dissemination of tumor cells from the primary site to distant organs that they will colonize [11]. 

Different molecular PDAC subtypes were identified from transcriptional profiling of resected PDAC tumors with the purpose of predicting disease progression and exploring the possibility to direct the patient to a personalized treatment. Among these subtypes, pancreatic progenitor or classical, squamous or quasi-mesenchymal/basal-like and exocrine-like PDAC were defined [32,33,34,72]. While the pancreatic progenitor subtype is well-to-moderately differentiated and shows an enrichment of endodermal markers with a slightly better prognosis [34], squamous PDAC are poorly differentiated, highly chemo-resistant [73] and lead to a worse outcome [34]. These two subtypes can possibly co-exist [74] conferring a higher plasticity [75,76]. The analysis of data merged from the studies by Collisson et al., Moffitt et al. and Bailey et al. comprising a data set from the International Cancer Genome Consortium (ICGC) and The Cancer Genome Atlas (TCGA) cohorts was performed by Sivakumar. Therein, PDAC was categorized into three subtypes. The subtype 1, also called Hedgehog, was associated with quasi-mesenchymal, basal, activated stroma or squamous subtypes; the second subtype or NOTCH was associated with exocrine-like, normal stroma or ADEX subtypes; lastly, subtype 3 or Cell Cycle was associated with classical or pancreatic progenitor subtypes. Subtype 1, characterized by poor prognosis, is enriched for neutrophils and NK cells and is potentially a target for immunotherapy, including myeloid depletion therapy [77]. Later, Puleo as well reviewed previous transcriptional profiles in a large cohort, eventually identifying five subsets (i.e., pure classical, immune classical, desmoplastic, stroma-activated, and pure basal-like) consistently with the type of stromal infiltration [32].

Owed to its aggressive yet subtle nature, the disease remains mostly unnoticed till an advanced stage, so that in the majority of the cases the diagnosis is made when the cancer has already spread to other organs, representing the seventh cause of cancer-related mortality [78]. PDAC being rather uncommon (crude rate, 6.0 per 100,000) [78], the screening of asymptomatic general population aimed at its early diagnosis is inappropriate, as the amount of false positive results generated even by a highly specific (i.e., at 99% value) test would overcome the number of true positive ones. Accordingly, rather than a screening approach, the surveillance of individuals in high-risk groups is currently considered a priority [79,80]. Such an approach has been referred to as DEF: define, enrich and find [80]. Along this line, the U.S. preventive services Task Force and others have identified such high-risk groups. Subjects at high-risk of developing PDAC comprise individuals with new-onset diabetes mellitus [81,82,83,84,85,86,87], those with at least two first-degree relatives with PDAC, patients with (hereditary) chronic pancreatitis or defined predisposition syndromes (i.e., Peutz–Jeghers syndrome, those with germline pathogenic variants in *CDKN2A*, *BRCA2*, *PALB2*, or in the genes of the DNA mismatch repair system associated with Lynch syndrome) [88,89], or individuals with intraductal papillary mucinous neoplasms (IPMN) [90,91,92,93].

So far, the diagnosis of PDAC relies on computed tomography imaging and EUS accompanied/confirmed by cyto/histological specimens. Currently, no established molecular tool exists for PDAC diagnosis besides carbohydrate antigen (CA19.9), with its long-lasting pros and cons [94]. Recently, this marker has received new attention, as a study has shown that its levels measured over time by a plex enzyme-linked immunosorbent assay to decrease nonspecific cross-reactivity can serve as an anchor marker for the early detection of PDAC (with sensitivity up to 50% at 99% specificity within 0–6 months before diagnosis for early-stage disease) [95]. The performances reached by measuring CA19.9 could be implemented by adding other protein markers, such as TIMP and LRG1 or THBS2 [95,96]. In parallel, it has also been shown that high levels of MUC5C circulating in extracellular vesicles act as biomarkers for the presence of invasive carcinoma within IPMN [97]. However, the one-size-fits-all approach in looking for suitable PDAC biomarkers has been paralleled by approaches looking for multiple alterations, as done by targeting *KRAS* (and other) mutations in circulating tumor DNA coupled with the assessment of thresholded proteins [98,99]. Similarly, also targeting epigenetic alterations of circulating DNA could allow detecting PDAC [100,101], as did the investigation of as many as 29 protein biomarkers [102].

To date, the possibility of detecting tumor cells or their derivatives in the circulation is being explored in clinical practice for the diagnosis of solid tumors and is also suitable for the management of neo-adjuvant therapy [103].

Noticeably, having shown that CTCs are detectable in the circulation in advance of PDAC formation in GEMM [2], Rhim and coll. also detected pancreas epithelial cells in the blood of patients with both pancreatic cystic lesions and no clinical evidence of cancer and of patients with pancreatic cancer [104]. This report strongly points to the fact that cancer cells could be detectable even before the detection of cancer.

CTCs are thought to be heterogeneous groups of cells with varying phenotypic and genotypic properties. However, the identification of CTCs that underwent/are going through EMT and their impact on current detection of PDAC remains poorly investigated, although CTC analysis in PDAC patients supports the association of EMT features with portal vein invasion and lymph node metastasis [105,106].

CTC enrichment strategies which target epithelial markers can reduce the uptake of recovered CTC, failing to intercept all possible forms of disseminated tumor cells [2,107]. An improvement would be possible by the introduction of potential mesenchymal targets including *ZEB1*, SNAI1 (SNAIL), Vimentin, N-Cadherin, FGFR2, PLS3, *Twist1* and PI3K/AKT [108].

The presence of mesenchymal EMT markers such as *ZEB1* (Zinc finger homebox 1) and the expression of epithelial CK (cytokeratin) in PDAC have shown no statistically significant impact on survival in patients with metastatic PDAC [109]. The overexpression of the EMT inducer MUC-1 in CTCs was associated with a median overall survival of patients nearly three times shorter than that of those with MUC-1 negative CTCs [110].

Two recent studies evaluated the potential impact of three subpopulations CTCs undergoing EMT in PDAC patients. Using the CanPatrol system, Zhao et al. captured and characterized the CTCs by means of four epithelial biomarkers, specifically the epithelial cell adhesion molecule (EpCAM) and three types of cytokeratins (named CK 8, 18 and 19), together with two mesenchymal biomarkers, likely vimentin and twist, plus the leukocyte biomarker CD45. They defined three CTC subgroups: epithelial (E-CTC), mesenchymal (M-CTC), plus a hybrid population expressing both epithelial- and mesenchymal-specific genes (E/M-CTC). The presence of CTCs correlated with poor patient prognosis and with lymph node metastasis, advanced TNM stage, lower blood lymphocyte counts, and higher neutrophil-to-lymphocyte ratio [111]. Only M-CTCs were present in significantly more patients with distant metastasis than in those without, and positively correlated with TNM stage (being more abundant in stage III and IV) [112]. Semaan et al. utilized di-electrophoresis-field flow fractionation (DEP-FFF) to isolate CTCs and by multiplex imaging flow cytometry revealed four distinct sub-populations of CTCs: E-CTC (EpCAM and/or Pan-CK positive), M-CTC (VIM positive), partial epithelial–mesenchymal transition (pEMT-CTC—positive to epithelial and mesenchymal antibodies) and stem cell-like (SC-CTC—CD133 positive). Although total number of CTCs did not correlate with clinicopathological variables, pEMT-CTCs correlated with advanced disease, worse progression-free and overall survival in all patients, and earlier recurrence after resection [112].

Another approach to evaluating the presence of CTCs in peripheral blood of PDAC patients is presented in a clinical trial (NCT04323917) covered by patent (No. EP13197367) and measuring the levels of EMT-Transcription factor (EMT-TFs) mRNAs with the primary aim to depict the molecular profile of EMT-TFs variations in the blood of patients with early, intermediate or advanced PDAC with respect to disease progression. Secondarily, the researchers aspire to identify biomarkers suitable for the selection of patients amenable of responsiveness to medical and surgical treatment [113,114].

The great variability of methods used for the isolation of CTCs by epithelial or mesenchymal or mixed epithelial and mesenchymal markers associated with the prediction of a better or worse prognosis does not make the data unambiguous and clear [107,115,116,117,118,119,120,121,122,123,124,125,126,127,128,129,130]. It would therefore be necessary to integrate more tests for the complete evaluation of EMT in CTCs and their correlation with disease behavior and progression to improve patient management in the clinical arena (Figure 2).

### 3.2. EMT, Chemoresistance and Choice of PDAC Treatment

The responsiveness to drug agents resulting in improved or poor prognosis of PDAC patients is a multifactorial pathway in which even EMT process is involved. Albeit gemcitabine (GEM) confers a more favorable survival, strengthening its role as first-line adjuvant therapy respect to 5-fluorouracil (5-FU) [131,132], EMT plasticity confers resistance to this drug in both human pancreatic cancer cells [133,134] and in mouse models [46]. On the other hand, the EMT suppression in primary tumor does not alter the emergence of invasive PDAC, but contributes to enhanced sensitivity to GEM treatment and, consequently, to a better overall survival of mice [46].

In the same manner, intervening into the increase of miR-33a expression as a tumor suppressor at the cellular level with consequent downregulation of β-catenin, EMT-TFs (i.e., slug, vimentin, and N-cadherin) and of survivin, cyclin D1, and MDR-1 expression could mediate a greater sensitivity to GEM [135].

Another important transcription factor involved in cell differentiation activating or suppressing gene expression is GATA-6 [136,137]. It has a direct role though the regulation of epithelial and mesenchymal genes. In patients affected by basal-like subtype of pancreatic cancer, the overall survival is lower than classical subtype (10% vs. 33%), such as the progression rate is higher (60% vs. 15%) [138].

Among different extracellular mediators and intracellular pathways, TGF-β, IL-6 and -1 and Hedgehog subtype are considered the major drug targets to inhibit EMT by using direct antibodies or inhibitors of their signaling [139,140].

## 4. Conclusions

Despite the studies carried out so far and the advancement in accepting the notion of EMT as a hallmark of PDAC, the network of mechanisms regulating its timing and regulation requires further research. The results reported in organoid and mouse model studies underline the way in which genome editing contributed to better understanding of the phenomenon. Although this is customary in cellular and animal models, it is obviously not foreseeable in humans.

Currently, the identification by translational studies of EMT markers suitable for deciphering the aggressive behavior of PDAC could possibly contribute to the advancement in diagnosis and monitoring of its evolution and of responsiveness to treatments.

## Figures and Tables

**Figure 1 cancers-14-05797-f001:**
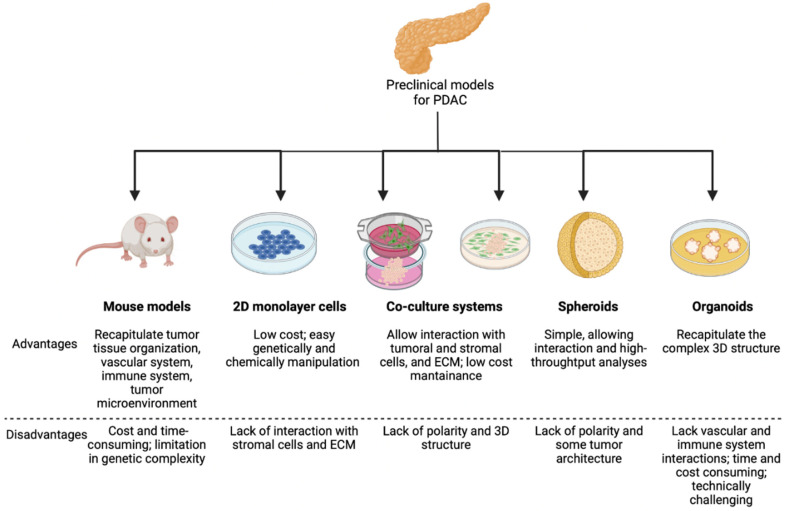
Scheme of the available preclinical models for PDAC studies, with their advantages and limitations. Created with BioRender.com (accessed on 10 June 2022).

**Figure 2 cancers-14-05797-f002:**
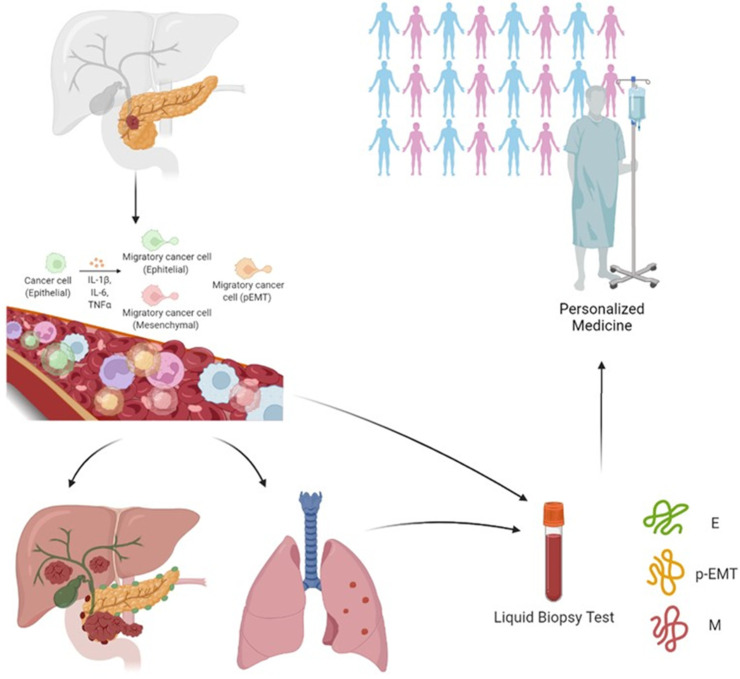
The presence in the blood of EMT markers from pancreatic cancer migratory cell (CTC), epithelial (E-CTC), mesenchymal (M-CTC) and/or hybrid population expressing both epithelial- and mesenchymal-specific genes (E/M-CTC or partial EMT cells) could represent great potential for anticipating PDAC detection, possibly providing a new tool for personalized medicine. Created with BioRender.com (accessed on 10 June 2022).

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
