# Peer review of "Epithelial to Mesenchymal Transition as Mechanism of Progression of Pancreatic Cancer: From Mice to Men"

_cancers, 2022, doi:10.3390/cancers14235797_

Round 1
Reviewer 1 Report
The present review addresses an important topic regarding pancreatic cancer biology that is epithelial to mesenchimal transition (EMT). The authors offer an exhaustive description of the principal current models for in vitro and in vivo study of pancreatic cancer development, with particular attention to the analysis of EMT molecular pathways. They also mentioned interesting studies regarding protein biomarkers involved in pancreatic cancer diagnosis and subpopulations of circulating tumor cells correlating with patients prognosis. They don't discuss the relationship between EMT and cancer stem cells which are considered strictly associated in the recent literature. I was wondering what the Authors think about this and if they consider appropriate to include a paragraph on this subject in the present review.
Author Response
The present review addresses an important topic regarding pancreatic cancer biology that is epithelial to mesenchimal transition (EMT). The authors offer an exhaustive description of the principal current models for in vitro and in vivo study of pancreatic cancer development, with particular attention to the analysis of EMT molecular pathways. They also mentioned interesting studies regarding protein biomarkers involved in pancreatic cancer diagnosis and subpopulations of circulating tumor cells correlating with patients prognosis. They don't discuss the relationship between EMT and cancer stem cells which are considered strictly associated in the recent literature. I was wondering what the Authors think about this and if they consider appropriate to include a paragraph on this subject in the present review.
R.: We thank the reviewer for the note. Although we recognize the role of EMT in CSCs, we have limited ourselves to describing a small paragraph (lines 63-75) in the introduction in which we highlight that this phenomenon is not only typical of CSCs but that even tumor cells or in the process of tumor transformation acquire typical characteristics of CSCs in the process of growth and invasion. For more details, please refer to our previous review on EMT and Gastrointestinal cancer: “Greco L, Rubbino F, Morelli A, Gaiani F, Grizzi F, de'Angelis GL, Malesci A, Laghi L. Epithelial to Mesenchymal Transition: A Challenging Playground for Translational Research. Current Models and Focus on TWIST1 Relevance and Gastrointestinal Cancers. Int J Mol Sci. 2021 Oct 25;22(21):11469. doi: 10.3390/ijms222111469. PMID: 34768901; PMCID: PMC8584071.”
"Please see the attachment."

Reviewer 2 Report
In this manuscript, the authors discuss the role of EMT in pancreatic cancer, covering several preclinical models used in the study of pancreatic cancer, as well as subtypes, and CTCs. The manuscript is in general well written, however some sections are lightly touched upon. Given space limitations, and the various topics the authors want to include, I would suggest they focus on aspects related to EMT.
Major points:
1. In general, EMT confers cells with migrating and invasive properties, but many papers have described the role of EMT and EMT induces in additional processes of tumour biology (stemness, survival, therapy resistance…). These concepts should be integrated in the corresponding chapters.
2. Genetically engineered mouse models.
I find the authors discuss a limited number of models and publications. For example, ref. 5, though included in the manuscript, is not discussed in this chapter. Additional, more recent literature that could be included is the role of SNAI2 (for example Recouvreux et al., J. Exp. Med. 2020 ), SNAI1 and 2 (Lan et al., Nature 2022)… amongst others. In fact, the implication of EMT/EMT inducers in metastasis is a very active debate in pancreatic cancer.
3. Heterotypic human cell co-culture models.
This part seems to be a bit disconnected to EMT.
4. Spheroids
EMT and EMT inducers additionally confer cells with “stemness” properties.
I did not fully understand the paragraph between lines 196 and 203. Since there is no reference in this paragraph, I could not check the literature to clarify the point the authors wanted to make.
5. Organoids
This chapter seems rather brief (as the two previous ones), given the extensive literature dedicated to this topic. I do not see a clear link to the study of EMT.
6. EMT and human pancreatic cancer
In lines 77-86, the concept of the different subtypes in PDAC is introduced. Some information is repeated (though more elaborated) in this chapter. The authors focus on transcriptomic analysis and do not mention other types of analysis used to stratify. Additionally, PDAC is not homogeneous, the existence of intratumour heterogeneity has been described. The authors summarize in lines 253-265 the relation of EMT with prognosis, but which is the relation of the different subtypes with EMT?. The topic of PDAC subtypes has been the subject of several reviews, I would suggest the authors to focus on its relation to EMT.
When describing CTCs, a conclusion is not clear (if EMT promotes CTC), as what the different signatures mean (pEMT in one study correlates with advanced disease, whereas M-CTC in another study, if I understood correctly). There is additional literature the authors should refer in this part.
7. Conclusions.
The authors write “the review highlights how the identification of EMT markers could contribute to (…) responsiveness to treatments”. Neither therapies for PDAC nor how EMT promotes therapy resistance are discussed in this review. This is an important topic that takes into account subtypes, CTC and other aspects of PDAC progression, and should be included in the manuscript.
Minor points:
Line 54: I wonder whether “metamorphosis” is the right word.
Line 71: “inactivation of TP53”. I understood TP53 loss by this, whereas missense mutations are the most common event on TP53 in PDAC.
Line 99: There seems to be a typo in “Pdx2” (should be Pdx1, according to ref. 28).
The sentence of 163-164 “the human equivalent of murine strong EMT has not been described” should be rephrased for clarity.
Lines 1001-105: Please check the references. Ref 32 includes additional alleles (p16/p19).
Some references seem to be missing. For example, at the end of the sentence in lines 195, 139, 203.
Author Response
Comments and Suggestions for Authors
In this manuscript, the authors discuss the role of EMT in pancreatic cancer, covering several preclinical models used in the study of pancreatic cancer, as well as subtypes, and CTCs. The manuscript is in general well written, however some sections are lightly touched upon. Given space limitations, and the various topics the authors want to include, I would suggest they focus on aspects related to EMT.
Major points:
- In general, EMT confers cells with migrating and invasive properties, but many papers have described the role of EMT and EMT induces in additional processes of tumour biology (stemness, survival, therapy resistance…). These concepts should be integrated in the corresponding chapters.
R.: We thank the reviewer for the note, so we have included the suggested notions in the “Introduction” now entitled “Complexity of metastatic cascade, embracing stromal and mesenchymal traits, and the importance of its application to human pancreatic cancer”, new lines 63-75 of the revised copy, even though they are not the focus of our review.
- Genetically engineered mouse models.
I find the authors discuss a limited number of models and publications. For example, ref. 5, though included in the manuscript, is not discussed in this chapter. Additional, more recent literature that could be included is the role of SNAI2 (for example Recouvreux et al., J. Exp. Med. 2020 ), SNAI1 and 2 (Lan et al., Nature 2022)… amongst others. In fact, the implication of EMT/EMT inducers in metastasis is a very active debate in pancreatic cancer.
R.: We thank the reviewer for the suggestion, so we have included a paragraph in new lines 143-148, in lines 178-199 and in lines 218-226
- Heterotypic human cell co-culture models.
This part seems to be a bit disconnected to EMT.
R.: We thank the reviewer for the suggestion, we rephrased our sentence adding an explanatory sentence because in this cellular CAFs co-culture model is one of the useful pathways for EMT triggering
- Spheroids
EMT and EMT inducers additionally confer cells with “stemness” properties.
I did not fully understand the paragraph between lines 196 and 203. Since there is no reference in this paragraph, I could not check the literature to clarify the point the authors wanted to make.
R.: Lines 195 and 203, now included from new lines 250 - 278 refer to the same new ref 57. We apologize for the lack and we reported the reference at the end of the relative sentence.
We have also included two new references in the paragraph: Gagliano et al 2016 (lines 254-258) and Bradney et al., 2020 (lines 279-290).
- Organoids
This chapter seems rather brief (as the two previous ones), given the extensive literature dedicated to this topic. I do not see a clear link to the study of EMT.
R.: The chapter on organoids is actually a subsection of Chapter 2. We thank the reviewer for the suggestion but the literature available to date on organoids in PDAC is so much but, at the same time, only the works reported relate organoids, PDAC and EMT.
- EMT and human pancreatic cancer
In lines 77-86, the concept of the different subtypes in PDAC is introduced. Some information is repeated (though more elaborated) in this chapter. The authors focus on transcriptomic analysis and do not mention other types of analysis used to stratify.
R.: We thank the reviewer for the note. Only Bailey's work is on a genetic basis integrated with transcriptomics. The rest of the work for the molecular subtype is all based on transcriptomics/gene expression.
Additionally, PDAC is not homogeneous, the existence of intratumour heterogeneity has been described. The authors summarize in lines 253-265 the relation of EMT with prognosis, but which is the relation of the different subtypes with EMT?. The topic of PDAC subtypes has been the subject of several reviews, I would suggest the authors to focus on its relation to EMT.
R.: The focus on PDAC subtypes and relation to EMT is reported in the new lines 329-341.
When describing CTCs, a conclusion is not clear (if EMT promotes CTC), as what the different signatures mean (pEMT in one study correlates with advanced disease, whereas M-CTC in another study, if I understood correctly). There is additional literature the authors should refer in this part.
R.: We thank the reviewer for the note. We have included a paragraph according to this suggestion and moved Figure 2 to the end of this chapter.
- Conclusions.
The authors write “the review highlights how the identification of EMT markers could contribute to (…) responsiveness to treatments”. Neither therapies for PDAC nor how EMT promotes therapy resistance are discussed in this review. This is an important topic that takes into account subtypes, CTC and other aspects of PDAC progression, and should be included in the manuscript.
R.: We thank the reviewer for the note. We have included a new sub-paragraph in chapter 3 on “EMT, chemoresistance and choice of PDAC treatment” before the conclusion. In this regard, we have moved Figure 2 to the end of the new Chapter 3. In addition, we have revised the text of the conclusion.
Minor points:
Line 54: I wonder whether “metamorphosis” is the right word.
R.: Right, typo in “metamorphoses”
Line 71: “inactivation of TP53”. I understood TP53 loss by this, whereas missense mutations are the most common event on TP53 in PDAC.
R.: As reported in reference: “For TP53, nonsense mutations, frameshift mutations, and mutations predicted to deleteriously impact splice sites were also classified as non-functional. However, TP53 missense mutations were further classified as functional or non-functional according the experimental functional classification and prediction of dominant-negative activity as reported in the IARC TP53 somatic mutation database (http://www-p53.iarc.fr/). (ref Winter JM, Tang LH, Klimstra DS, et al. Failure patterns in resected pancreas adenocarcinoma: lack of predicted benefit to SMAD4 expression. Ann Surg. 2013;258(2):331-335.) If no SNVs were detected for a given gene in a case, it was assigned a functional status.”
Line 99: There seems to be a typo in “Pdx2” (should be Pdx1, according to ref. 28).
R.: We thank the reviewer for the note. Right, it is a typo in “Pdx2”, right word is “Pdx1”
The sentence of 163-164 “the human equivalent of murine strong EMT has not been described” should be rephrased for clarity.
R.: We thank the reviewer for the suggestion and we have rephrased sentence as: “such a strong EMT was only described in mice but not in human cells yet”
Lines 101-105: Please check the references. Ref 32 includes additional alleles (p16/p19).
Some references seem to be missing. For example, at the end of the sentence in lines 195, 139, 203.
R.: We thank the reviewer for the note. We apologize for the lack and we reported the reference at the end of the relative sentence.
"Please see the attachment."

Reviewer 3 Report
Dear authors,
Well done on data collection and including so many studies. However, I have some concerns/questions/remarks. Please see the following:
Introduction
This section is confusing as pancreatic cancer is only mentioned at the end of the introduction. this section needs more relevance e.g. why is this paper important for pancreatic cancer, what question do you aim to answer, will this have any benefits for treatment etc. In its current state I do not understand what your manuscript will be about.
Section 2:
Don't see the relevance of this section, not clear what it's supposed to tell.
Section 3:
- good review of the literature, but poor grammar makes it difficult to follow. Again, not sure what this paragraph is supposed to tell me and how it answers your research question/title.
- line 136: DPADC..?
- why only GEMMs? Why not also look at PDX models?
Section 4: can include more information.
Section 5: It took a while before realizing what this section was about. The main gist was only mentioned in the last sentence, but don't see the relevance of the rest of this section to research question.
Section 6: No relation to research question.
Section 7:
- Again, well done on data collection, but this section is very long winded. There is too much information on topics not related to your research question.
- Line 346-348 these seem like important findings, why wasn't this elaborated on more? These are the kind of things this section should be about as they make your story stronger. More important findings are mentioned after this too in later lines - why did it take so long to get here?
Conclusion: This isn't what is written in the manuscript at all. There is also a figure here which would have been of more relevance earlier on.
Overall: the authors have put in a lot of work in reviewing the literature, but overall, this manuscript fails at telling the story the title suggests. The abstract also does not match the manuscript whatsoever. The text is quite scattered, and it is unclear where the manuscript is going. Would have also been nicer to see an overview of the regulatory network of EMT and where key players are involved. Either change the title of the work or add more structure. This work is also not new, there are many publications of EMT in PDAC. Last, the use of English is poor. There are grammatical errors, and the text reads like the most complicated synonym was used for every word making it impossible to follow at times.
Author Response
Comments and Suggestions for Authors
Dear authors,
Well done on data collection and including so many studies. However, I have some concerns/questions/remarks. Please see the following:
Introduction
This section is confusing as pancreatic cancer is only mentioned at the end of the introduction. this section needs more relevance e.g. why is this paper important for pancreatic cancer, what question do you aim to answer, will this have any benefits for treatment etc. In its current state I do not understand what your manuscript will be about.
R.: We thank the reviewer for his comments and suggestions, following which we have allowed ourselves to revise our manuscript trying to give a greater logical thread.
Section 2:
Don't see the relevance of this section, not clear what it's supposed to tell.
R.: We thank the reviewer, so we clarify his doubt. We attempt to resume the pivotal, yet largely unraveled, role of EMT in the progression of pancreatic cancer because our understanding of the relationship between tumor microenvironment and EMT is not yet mature and needs additional data from basic, translational, and clinical sciences for clarification. Animal models strongly point to EMT as a key feature for PDAC dissemination, and we will briefly cover these preclinical models, which are listed in text and schematized in Figure 1.
Section 3:
- good review of the literature, but poor grammar makes it difficult to follow. Again, not sure what this paragraph is supposed to tell me and how it answers your research question/title.
R.: We thank the reviewer. We have pointed out to editor that paragraphs 3-6 are subparagraphs of chapter 2, so section 3 represents one of the main PDAC models.
- line 136: DPADC..?
R.: Right, typo in “DPADC”, right word is “PDAC”
- why only GEMMs? Why not also look at PDX models?
R.: We thank the reviewer. We pointed out only on GEMMs because PDX model is generally used to study PDAC and not EMT in PDAC, as it happens in GEMMs.
Section 4: can include more information.
R.: We thank the reviewer. We have pointed out to editor that paragraphs 3-6 are subparagraphs of chapter 2, so section 4, includes the most relevant literature about PDAC and EMT.
Section 5: It took a while before realizing what this section was about. The main gist was only mentioned in the last sentence, but don't see the relevance of the rest of this section to research question.
R.: We thank the reviewer. We have pointed out to editor that paragraphs 3-6 are sub-paragraphs of chapter 2, so section 5 represents one of the PDAC models. We have also included two new references in the paragraph: Gagliano et al 2016 (lines 254-258) and Bradney et al., 2020 (lines 279-290).
Section 6: No relation to research question.
R.: We thank the reviewer. We have pointed out to editor that paragraphs 3-6 are subparagraphs of Chapter 2, so section 6 represents one of the PDAC models related to EMT.
Section 7:
- Again, well done on data collection, but this section is very long winded. There is too much information on topics not related to your research question.
R.: We thank the reviewer, so we clarify his doubt. As mentioned in the title of paragraph we depicted the current knowledge of translational studies focused on EMT and PDAC.
- Line 346-348 these seem like important findings, why wasn't this elaborated on more? These are the kind of things this section should be about as they make your story stronger. More important findings are mentioned after this too in later lines - why did it take so long to get here?
R.: We thank the reviewer for the note. We have included a paragraph at the end of chapter according to this suggestion in new lines 444-456.
Conclusion: This isn't what is written in the manuscript at all. There is also a figure here which would have been of more relevance earlier on.
R.: We thank the reviewer for the note. We have included a new sub-paragraph in chapter 3 on “EMT, chemoresistance and choice of PDAC treatment” before the conclusion. In this regard, we have moved Figure 2 to the end of the new Chapter 3. In addition, we have revised the text of the conclusion.
Overall: the authors have put in a lot of work in reviewing the literature, but overall, this manuscript fails at telling the story the title suggests. The abstract also does not match the manuscript whatsoever. The text is quite scattered, and it is unclear where the manuscript is going. Would have also been nicer to see an overview of the regulatory network of EMT and where key players are involved. Either change the title of the work or add more structure. This work is also not new, there are many publications of EMT in PDAC. Last, the use of English is poor. There are grammatical errors, and the text reads like the most complicated synonym was used for every word making it impossible to follow at times.
R.: We thank the reviewer for his opinion to which we try to give an answer.
The segmentation found by him could be due to the incorrect layout of the subsections relating to chapter 2.
Our aim is not to highlight an overview of the regulatory network of EMT, but to synthesize current knowledge of EMT in relation to PDAC. We proposed an excursus from the in vitro model, passing through the animal model up to translational studies on the importance of EMT in clinical practice, opening new questions to the reader.
"Please see the attachment."
Round 2
Reviewer 2 Report
In this revised version, the authors have addressed the points brought up and updated the literature.
There are no further comments.